# *Annona crassiflora* Mart. Fruit Peel Polyphenols Preserve Cardiac Antioxidant Defense and Reduce Oxidative Damage in Hyperlipidemic Mice

**DOI:** 10.3390/foods12112097

**Published:** 2023-05-23

**Authors:** Eliana Akemi Komino, Letícia Pereira Afonso Ramos, Adriele Vieira de Souza, Douglas Carvalho Caixeta, Vinicius Prado Bittar, Ana Luiza Borges, Françoise Vasconcelos Botelho, Foued Salmen Espindola, Allisson Benatti Justino

**Affiliations:** 1Instituto de Biotecnologia, Universidade Federal de Uberlândia, Rua Acre s/n, Bloco 2E, Uberlândia 38400-319, MG, Brazil; komino.ea@gmail.com (E.A.K.); leticiapafonso@gmail.com (L.P.A.R.); adriele_vds@hotmail.com (A.V.d.S.); caixetadoug@gmail.com (D.C.C.); viniciusp.bittar@gmail.com (V.P.B.); analuizaborges.sb@gmail.com (A.L.B.); francoise.botelho@ufu.br (F.V.B.); fouedespindola@gmail.com (F.S.E.); 2Programa de Pós-Graduação da Rede Nordeste de Biotecnologia (RENORBIO), Universidade Federal de Alagoas, Maceió 57072-900, AL, Brazil

**Keywords:** araticum, cardiovascular disease, dyslipidemia, glutathione defense, oxidative stress

## Abstract

Dyslipidemia and oxidative stress are directly related to the pathogenesis of cardiovascular diseases. *Annona crassiflora* Mart. (ACM) has been traditionally used in folk medicine to alleviate inflammation and pain. This plant is rich in polyphenols, which exhibit high antioxidant capacity. The present study aimed to elucidate the antioxidant properties of ACM in the heart of hyperlipidemic mice. The animals were orally administered either a crude ethanol extract (CEAc) or a polyphenols-rich fraction (PFAc) obtained from ACM fruit peel. There were correlations between blood and fecal biochemical data with cardiac oxidative stress biomarkers. Here, the pre-treatment with CEAc for 12 d led to an increase in glutathione content (GSH) and a reduction in the activities of superoxide dismutase (SOD), catalase (CAT) and glutathione peroxidase. Moreover, PFAc was found to enhance the total antioxidant capacity as well as GSH, SOD and CAT activities, which were reduced by Triton WR-1339-induced hyperlipidemia. Moreover, the administration of PFAc before the treatment resulted in a decrease in protein carbonylation and lipid peroxidation levels, as well as a reduction in the activities of glutathione reductase and glucose-6-phosphate dehydrogenase. ACM fruit peel showed improvement in the glutathione system, mainly its polyphenols-rich fraction, indicating a potential cardioprotective antioxidant usage of this plant extract.

## 1. Introduction

Oxidative stress significantly impacts the progression of cardiovascular disease (CVD), which encompasses inflammation and tissue damage and is considered one of the leading causes of death globally [1,2,3]. There is considerable proof that high cholesterol levels cause oxidative stress (OS) in the heart, contributing to the development of atherosclerosis, hypertension, and myocardial ischemia-reperfusion injury [4]. OS is a physiological state caused by an imbalance in the body’s reactive oxygen species (ROS) and antioxidants. ROS are produced as a byproduct of regular metabolism, and their buildup can cause cellular biomolecule damage and contribute to the etiology of many diseases [2]. ROS have the ability to cause oxidative damage to cell membranes, lipids and proteins, resulting in cellular malfunction and apoptosis [5]. Furthermore, ROS can impair nitric oxide bioavailability, induce endothelial dysfunction and hypertension [2,3].

On the other hand, there is a built-in antioxidant defense mechanism that can neutralize ROS and prevent oxidative damage [5]. Glutathione is one of the most significant antioxidants in the body, and it is essential for cellular redox equilibrium [5]. Many treatment options for oxidative stress and related disorders have been explored. To scavenge ROS and protect cells from oxidative damage, antioxidants such as vitamin C and E have been utilized [6]. Furthermore, therapeutic techniques that target oxidative stress enzymatic pathways such as glutathione reductase have been proposed as a strategy to manage redox homeostasis [5].

Polyphenol-based phytotherapy has shown promise in the prevention and treatment of cardiovascular disorders [2,7]. Polyphenols exhibit antioxidant, anti-inflammatory, antiplatelet and anti-hypertensive activities that can reduce OS and inflammation while ameliorating established mitochondrial, endothelial and cardiovascular dysfunction [3,7]. Polyphenols have been found to activate nuclear factor-erythroid 2-related factor 2 (Nrf2), a transcription factor that affects the expression of antioxidant and detoxifying enzymes [2], as well as to stimulate glutathione biosynthesis [8]. Moreover, polyphenols can inhibit nicotinamide adenine dinucleotide phosphate (NADPH) oxidase, a key generator of ROS in the vasculature [3].

*Annona crassiflora* Mart. (ACM) is native to Cerrado Biome widely used to treat inflammation and pain in folk medicine [9]. Various components of the plant, including the peel, leaf, fruit, and seed, have been utilized to treat a variety of ailments such as malaria, microbial infections, snakebites, venereal diseases, diarrhea, and as agents to prevent cancer [9]. The extract of ACM has been demonstrated to contain various polyphenols, including derivatives of quercetin, (epi)catechin, procyanidins, feruloyl-hexosides, and caffeoyl-hexosides, which are responsible for its greater antioxidant potential and reduced cytotoxicity [10]. Previously, we showed that a polyphenols-rich fraction from the peel of *A. crassiflora* fruit decreased lipid and protein oxidative damage and preserved the glutathione defense in the liver tissue of mice with Triton WR-1339-induced hyperlipidemia [11]. However, no studies are evaluating ACM protective effects on cardiac tissue. Thus, the present study investigated whether the crude extract of *A. crassiflora* fruit peel and its polyphenols-rich fraction are capable of reducing lipid and protein damage in the heart. Additionally, the study aimed to verify whether pretreatments with these fractions preserved the cardiac antioxidant enzymes of hyperlipidemic mice in a tyloxapol-induced acute dyslipidemia model.

## 2. Material and Methods

### 2.1. Preparation and Extraction of ACM Polyphenols

Access to genetic heritage components (reference No. 010743/2015-4) was authorized by the National Council for Scientific and Technological Development (CNPq), and the fruit peels of ACM were collected from natural resources in the north of Minas Gerais State, Brazil. Dr André Vito Scatigna (Federal University of Uberlandia) carried out species identification. A voucher specimen (HU-FU68467) was deposited in the herbarium of this institution. The peels were dehydrated and then assessed to weigh 1.0 kg before being subjected to the maceration technique for 3 d at 25 °C, in the absence of light, using 6 L of 98% ethanol (EtOH). The resulting solution was filtered and the EtOH was eliminated using a rotary evaporator (Buchi R-210, Flawil, Switzerland) at 40 °C under low pressure. This process was repeated three times. The freeze-dried crude ethanol extract (CEAc) weighing 30.0 g was dissolved in a mixture of methanol and water (9:1) in a volume of 225.0 mL. This solution was then subjected to a liquid–liquid partitioning procedure following the method described by Justino AB et al. [10]. Briefly, using a separation funnel, liquid–liquid partition was performed with solvents of increasing polarity: *n*-hexane, dichloromethane, ethyl acetate and *n*-butanol. For this process, four extractions were performed with approximately 200.0 mL of each solvent. The rotary evaporation process was utilized at 40 °C to eliminate all solvents present in the fractions while operating under low pressure. The resulting fractions were frozen, lyophilized, and stored at −20 °C. For the experiments, CEAc and PFAc, fractions abundant in polyphenols obtained from ethyl acetate, were utilized. The analytical grade solvents used for the extraction and fractionation of the plant (methanol and ethyl acetate) were purchased from Sigma (Sigma, St. Louis, MO, USA) with an HPLC purity grade greater than 99.9% according to the following specifications: ≤0.0005% non-volatile matter, ≤1 ppb fluorescence (quinine) at 254 nm, ≤1 ppb fluorescence (quinine) at 365 nm, ≤0.005% free acid (as CH_3_COOH), and <0.03% water (Karl Fischer), with a maximum residue of 10 ppm after evaporation.

### 2.2. Phytochemical Prospection

The HPLC-ESI-MS/MS technique was used to identify the main compounds in CEAc and PF-Ac. The samples were dissolved in a mixture of water and isopropanol and then infused using a syringe with a pump flow of 200 μ/h. The column was heated to 200 °C, with a nebulizer gas flow of 4 L/min and 4.5 kV. The electron impact energy ranged from 5 to 45 eV. The HPLC conditions included a Shimadzu Shim-pack XR-ODS III column with a particle size of 5 µm and a pore diameter of 110 Å. The mobile phase consisted of water acidified with formic acid (0.1% *v*/*v*) and acetonitrile acidified with formic acid (0.1% *v*/*v*), and the gradient solvent system was as follows: 15–40% (0–7 min); 40–90% (7–8 min); 90–90% (8–9 min); 90–15% (9–10 min). The flow rate was 0.35 mL/min, and detection was performed at UV wavelengths of 280 and 360 nm. The data were collected in negative mode and covered a range of 20–1000 m/z.

To quantify the overall phenolic content, CEAc and PFAc were mixed with methanol (500 µg/mL) and then combined with a Folin–Ciocalteu solution (10% *v*/*v*) and a Na_2_CO_3_ solution (7.5% *m*/*v*). The mixture was heated to 50 °C for 5 min and the absorbance was recorded at 760 nm using a Thermo Scientific Genesys 10S UV–Vis spectrophotometer. The amount of total phenolic content was determined by comparing the results to a calibration curve that was made using gallic acid as the standard. To quantify the proanthocyanidin content, CEAc and PFAc were diluted in methanol (100 µg/mL) and then combined with a vanillin (0.5% *m*/*v*) in H_2_SO_4_ solution (70% *m*/*v*). The mixture was incubated at 50 °C for 15 min and the absorbance was measured at 500 nm (Thermo Scientific Genesys 10S, Thermo Fisher Scientific, Waltham, MA, USA). The proanthocyanidin content was calculated by comparing the results to a calibration curve that was created using epicatechin as the standard. The tests were conducted in triplicate, and the results were reported as milligrams of gallic acid equivalents (GAE/g) and milligrams of epicatechin equivalents per gram (EE/g), respectively [10].

### 2.3. Animals and Experimental Design

The material of this study consists of mice hearts under acute hyperlipidemia induced by tyloxapol. Hearts were collected after euthanasia and frozen in liquid nitrogen and subsequently stored at −80 °C. The animal experiments followed the ARRIVE guidelines and adhered to the National Institutes of Health Guide for the Care and Use of Laboratory Animals (NIH Publication 8023, revised 1978); as the experiments also received approval from the Ethics Committee (CEUA/UFU, protocol 039/19). CBEA/UFU provided 50 male mice of the C57BL/6 lineage, four weeks old. Initially, these animals were weighed and distributed in ten cages with dimensions of 30 × 19 × 13 cm. Each cage held five animals, all of which were housed in the CBEA, under consistent condition (12 h light/dark cycles, temperature: 23–25 °C), with free access to food and water.

The mice were divided randomly into eight groups for the study: the naïve (normolipidemic control) group, the hyperlipidemic (HLc) vehicle group, and the HLc group receiving oral administration of 10, 30, or 100 mg/kg of CEAC or PFAc daily for 12 days. The CEAc or PFAc was diluted in filtered water and administered daily for 12 consecutive days, while the naïve and HLc vehicle groups received only filtered water. On the thirteenth day, except for the mice in the naïve group, an intraperitoneal injection of 400 mg/kg of Triton WR-1339 (tyloxapol) was administered to induce a dyslipidemic condition [12]. The treated groups received the final oral administration of CEAc or PFAc 24 h after the tyloxapol injection. The animals were euthanized intraperitoneally with a combination of ketamine (80 mg/kg) and xylazine (10 mg/kg), diluted in sodium phosphate buffer with a pH of 7.3, 48 h after the tyloxapol administration.

### 2.4. Tissue Preparation

First, to prepare the heart homogenates (HHs), a 10 mL solution of phosphate–saline (PBS) was used to perfuse the mice. The hearts were quickly removed and frozen in an ultrafreezer (−80 °C) until the status redox analysis. The hearts were defrosted, thawed, weighed and homogenized in a phosphate buffer (at a ratio of 1:10 *w*/*v*, pH 7.4) for the assessment of OS markers. The homogenized heart was subjected to centrifugation at 800× *g* for 10 min at 4 °C, and the amount of protein present in the supernatant was determined using the Bradford method [13].

### 2.5. Oxidative Stress Markers Analysis

#### 2.5.1. Analysis of Oxidative Damage

The TBARS assay, as described by Yagi K [14], was used to measure lipid peroxidation (LP). The TBARS method involves the reaction between malondialdehyde (MDA) and thiobarbituric acid reactive substances. MDA is one of the byproducts released when free radicals attack the lipid membrane. The LP was quantified by constructing an analytical curve with 1,1,3,3-tetrametroxypropane (fluorescence measured at 515 nm_ex_ and 553 nm_em_, PerkinElmer LS 55, Boston, MA, USA). The presence of carbonyls in proteins was determined using 2,4-dinitrophenylhydrazine (DNPH), which reacts with the carbonyl group to form dinitrophenyl (DNP). The resulting compound was measured spectrophotometrically at 370 nm using a PerkinElmer LS 55 instrument [15]. The antioxidant capacity was evaluated by the Ferric Reducing Antioxidant Power assay [16] at 593 nm (Molecular Devices) and using 6-hydroxy-2,5,7,8-tetramethylchroman-2-carboxylic acid to construct an analytical curve [11].

#### 2.5.2. Analysis of Enzymatic and Non-Enzymatic Antioxidants

Superoxide dismutase (SOD) activity was evaluated according to the autoxidation inhibition of pyrogallol by SOD present in HH [17]; the enzymatic kinetic was conducted in 96-well microplates at 420 nm for 10 min (Molecular Devices, San Jose, CA, USA) [18]. The catalase (CAT) activity was based upon hydroxide peroxide decomposition by CAT present in the heart samples [19]. The monitoring of hydrogen peroxide decomposition was conducted for a period of 10 min, in 96-well microplates, with the absorption being measured at 240 nm (PerkinElmer LS 55, PerkinElmer Inc., Waltham, MA, USA) [18]. To measure the amount of reduced glutathione (GSH), the proteins in the HHs were first extracted by adding metaphosphoric acid (1:1) and then centrifuging the mixture at 7000× *g* for 10 min. The resulting supernatants were then mixed with o-phthaldialdehyde (1 mg/mL), and the fluorescence was measured at an excitation wavelength of 350 nm and an emission wavelength of 420 nm in 96-well microplates (PerkinElmer LS 55); GSH content was determined according to an analytical curve constructed with GSH [11]. Glutathione peroxidase (GPx) activity was determined according to the nicotinamide adenine dinucleotide phosphate (NADPH) decay [20], which was registered for 10 min, in 96-well microplates, at 340 nm (Molecular Devices) [11]. Glutathione reductase (GR) activity was measured according to the consumption of NADPH by oxidized glutathione [20]; NADPH decay was monitored in 96-well microplates for a duration of 10 min, with the absorption being measured at 340 nm (Molecular Devices) [11]. The activity of glucose 6-phosphate dehydrogenase (G6PD) was determined in 96-well microplates using a kinetic assay that monitored the production of NADH at 340 nm (PerkinElmer LS 55) for 10 min [18].

### 2.6. Statistical Analysis

All analyses and graphs were performed in triplicate using the GraphPad Prism software (version 9.0). The results were expressed as mean ± standard error of the mean, and the one-way analysis of variance (ANOVA), followed by Dunnett’s post-test for multiple comparisons, were used to assess differences between the results. Pearson’s correlation was used to analyze linear relationships between the data considering a single value after euthanasia. Values of *p* < 0.05 were considered significant.

## 3. Results and Discussion

The high phenolic content in *A. crassiflora* Mart. (ACM) fruit peel, primarily consisting of proanthocyanidin polyphenols [10,21], makes it a potential source of cardioprotective agents due to its antioxidant properties. Our previous research demonstrated that the fruit peel of *A. crassiflora* is an exceptional source of polyphenols and alkaloids [10,21,22]. By performing liquid–liquid partitioning on the crude ethanol extract (CEAc) and utilizing HPLC-ESI-MS/MS analysis, we were able to concentrate and identify many proanthocyanidins, such as epicatechin and procyanidin B2, in the ethyl acetate fraction (referred to as the polyphenol-rich fraction from *A. crassiflora*, PFAc) (as depicted in Figure 1). These findings were also confirmed by previous studies conducted by Justino AB et al. [10] and Justino AB et al. [21], which reported the existence of quinic acid, epicatechin, quercetin–glucoside, and procyanidin B2 in CEAc and PFAc derived from ACM. It is noteworthy that the high-resolution spectra did not show the presence of possible metals derived from the solvents used for the fractionation.

The total phenolic content of CEAc and PFAc was determined to be 326 ± 2 and 535 ± 24 mg GAE/g, respectively. The quantification of proanthocyanidins was carried out specifically, as they are the major phenolic compounds present in PFAc. The liquid–liquid fractionation process effectively concentrated the proanthocyanidins in the PFAc fraction, resulting in a significantly higher content (767 ± 16 mg of EE/g) compared to CEAc (232 ± 5 mg of EE/g) (*p* < 0.001). Due to the availability of their phenolic hydrogens, proanthocyanidins have been discovered to produce persistent antioxidant-derived radicals and function as radical scavengers, singlet oxygen quenchers and electron donors [23]. In this context, we investigated the potential cardioprotective effects of the ACM fruit peel on redox balance indicators by evaluating the antioxidant properties of CEAc and PFAc in the hearts of hyperlipidemic mice.

To confirm the cardioprotective effects of ACM in vivo, an acute hyperlipidemia model was chosen using Triton WR-1339 (Tyloxapol, Tx), a lipoprotein lipase inhibitor [24]. The recently published data on plasma total cholesterol (TC) and triacylglycerol (TG) levels showed that pre-treatment with all doses of CEAc or PFAc for 12 days did not change TC and TG levels [11] (Appendix A). Moreover, TC and TG contents increased after intraperitoneal administration of 400 mg/kg of Tx (38 ± 4 to 359 ± 48 mg/dL, *p* < 0.001, and 64 ± 4 to 2990 ± 436 mg/dL, *p* < 0.01, respectively). After being induced with dyslipidemia and administered pre-treatment with either CEAc or PFAc, there were no discernible variances in the levels of plasma TC and TG compared to the vehicle group that received no treatment.

Following the administration of Tx, the mice that were not treated also displayed an increase in the overall lipid content in their feces, as evidenced by an increase from 13 ± 1 mg/g to 23 ± 2 mg/g (*p* < 0.01) (Appendix A). The groups that received pre-treatment with CEAc at dosages of 100 and 10 mg/kg demonstrated a difference in triglyceride content, with levels of 6.8 ± 0.2 mg/g observed in these groups compared to the vehicle group (7.7 ± 0.2 mg/g, *p* < 0.05) (Appendix A). In addition, after tyloxapol injection, pre-treatment with PFAc at a dosage of 30 mg/kg reduced cholesterol content, with levels of 5.4 ± 0.1 mg/g observed in the PFAc group compared to 5.9 ± 0.1 mg/g in the vehicle group (*p* < 0.01). These previous results validated the Tx-induced hyperlipidemia model.

Tyloxapol (Tx) was used to induce hyperlipidemia (HL) in the animals. Tx is a nonionic detergent that inhibits lipoprotein lipase and results in acute or chronic dyslipidemia. By enhancing the activity of 3-hydroxy-3-methylglutaryl coenzyme A reductase, an essential enzyme for cholesterol production, Tx encourages an increase in plasma triglycerides and cholesterol [12]. It is most likely because of the intraperitoneal delivery of Tx that the animals had greater amounts of TG and TC in their blood and feces.

Reduction in the total antioxidant capacity (TAC) (20% reduction in TAC, *p* < 0.05) and, consequently, increased lipid peroxidation (LP) (1.8 ± 0.0 vs. naive 0.9 ± 0.0 nmol TBA-RS/mg, *p* < 0.001) and protein carbonylation (PC) (21 ± 3 vs. naive 13 ± 1 nmol/mg, *p* < 0.05) were observed in the non-treated hyperlipidemic mice (Figure 2). Since lipids are particularly vulnerable to the attack of reactive oxygen and nitrogen species, it is well known that OS and increased free radical generation cause LP [25]. Proteins can also be damaged by ROS through oxidation, which leads to the generation of carbonyls and results in the depletion of thiol groups [26]. This is a sign of severe oxidative damage, which is associated with various pathologies, including cardiovascular diseases [27].

The findings demonstrated that PFAc dosages of 10, 30 and 100 mg/kg improved the cardiac total antioxidant capacity (TAC) (increase of over 45% in TAC, *p* < 0.01) (Figure 2a), while decreasing lipid peroxidation (LP) (1.2 ± 0.1 nmol TBA-RS/mg, *p* < 0.05) induced by Tx-induced hyperlipidemia (HL) (Figure 2b). PFAc at doses of 30 and 100 mg/kg were also able to reduce the protein carbonyl (PC) content (12 ± 1 nmol/mg, *p* < 0.05, Figure 2c). Earlier investigations have also shown that proanthocyanidins, which were found in higher levels in the PFAc, have the ability to decrease lipid peroxidation and protein carbonylation [11,21].

Additionally, this study contends that LP and HL caused by Tx-induced HL damaged the endogenous antioxidant profile. The defense system relies on antioxidant enzymes to limit the formation of ROS and repair OS-related damage [5]. The key enzymes involved in the process of removing ROS, particularly superoxide anion (O_2_^•-^) and H_2_O_2_, are SOD and catalase, respectively; SOD dismutes O_2_^•-^ forming H_2_O_2_; that is converted into H_2_O and O_2_ by CAT [28]. Here, an increase in superoxide dismutase (SOD) activity was observed in the untreated HLc mice (increase of 61%, *p* < 0.01, Figure 3a). Dyslipidemia involves increases in ROS such as O_2_^•-^ [28], which could have led to an overload of SOD. This overload may have contributed to the increased SOD activity observed in the vehicle group. As expected, the activity of CAT also increased in the vehicle (increase of 62%, *p* < 0.001, Figure 3b) in order to reduce the H_2_O_2_ formed through SOD pathway induced by dyslipidemia in the myocardium.

The pre-treatments of HLc animals with CEAc and PFAc resulted in the non-elevation of SOD (at 30 and 100 mg/kg for both CEAc and PFAc) and CAT (at 100 mg/kg for CEAc and at all doses for PFAc) activities (Figure 3a and Figure 3b, respectively). The possible antioxidant mechanism of these polyphenols-rich extracts was the reduction in ROS through the proanthocyanidins’ O_2_^•-^ scavenging properties [21,23].

With regard to antioxidant pathway of the glutathione system, the activities of GPx, GR and G6PD increased by over 45% in the untreated HLc mice (*p* < 0.01, Figure 4a–c, respectively), while the GSH content decreased (1.1 ± 0.1 vs. naive 1.7 ± 0.1 nmol/mg, *p* < 0.05, Figure 4d). Due to the observed increase in oxidative stress, as evidenced by the rise in LP and PC levels and reduction in total antioxidant capacity (Figure 2), GPx levels also increased. This increase in GPx activity may have led to a depletion of GSH in the hyperlipidemic animals, as GSH servers as a substrate for GPx during the detoxification of H_2_O_2_ [28]. This can be confirmed through the kinetic parameters obtained from GPx. The naive group had a Km of approximately 0.1 mM, while the hyperlipidemic group had a Km of approximately 0.029 mM. This indicates that GPx in hyperlipidemic animals has a greater affinity for GSH than non-hyperlipidemic animals. Therefore, it confirms glutathione depletion by increasing GPx activity.

The production of GSSG by GPx may lead to the back reduction of GSSG to GSH by GR, which uses NADPH as a reductant [29]. This observation explains the increased activity of GR observed in the cardiac tissue of the HLc animals (Figure 4b). The production of NADPH, which is essential for the regeneration of GSH by GR, relies on G6PD [28]. However, despite a higher activity of G6PD observed in the untreated HLc group, GSH content remained low, possibly due to its utilization as a means of controlling free radicals. GSH is the main soluble antioxidant that is present in significant concentrations in every cell compartment, which explains why TAC and the glutathione system are intimately connected, as shown in Figure 2d and Figure 4a, respectively [28]. All of these enzymatic and non-enzymatic antioxidant alterations induced by hyperlipidemia may have contributed to the increased lipid and protein damages observed in the vehicle animals.

The levels of GSH were higher in the groups treated with CEAc at 30 mg/kg and PAFc at 30 and 100 mg/kg, suggesting that these groups did not exhaust their endogenous antioxidant sources (Figure 4d). Furthermore, polyphenols may stimulate GSH synthesis [29]. These results corroborate the lower GR activity in the groups pre-treated with all PFAc doses (approximately 29 U/mg vs. vehicle 40 U/mg, *p* < 0.01), and the lower G6PD activity exhibited by the group pre-treated with 100 mg/kg of PFAc (23 ± 1 vs. vehicle 30 ± 2 U/mg, *p* < 0.05), since GSH levels were higher. Interestingly, among the pre-treatments, only CEAc at 100 mg/kg showed a reduction in GPx activity (decrease of 80%, *p* < 0.05, Figure 4a).

PFAc demonstrated significantly stronger antioxidant effects in comparison to the raw extract. The polyphenols-rich fraction reinstated the activities of SOD, CAT, GR, G6PD, and GSH contents in the hearts of HLc animals, resulting in the decrease in lipid peroxides and protein carbonyls. Additionally, it is worth noting that the highest PFAc dosage (100 mg/kg) was able to restore most of the oxidative and antioxidant parameters in the cardiac tissue of animals with tyloxapol-induced hyperlipidemia. The explanation for this could be that the bioactive compounds in the fraction do not exhibit these activities when present in lower concentrations.

Interestingly, the antioxidant response in the heart was distinct from that in the liver of mice with hyperlipidemia, as reported by Ramos LPA et al. [11]. Using the Tx-induced acute hyperlipidemia model, we observed that in cardiac tissue, the activities of SOD, CAT, GPx, GR, and G6PD increased, whereas in hepatic tissue, CAT and SOD did not change, and GPx, GR, and G6PD showed a decrease in their activities. This further highlights that the antioxidant system may respond differently in various organs and tissues of the same organism in an effort to better counteract oxidative damage.

Blood and fecal biochemical data, as well as OS biomarkers, were subjected to Pearson’s correlation with cardiac biomarkers of OS (Appendix A). It is worth noting that a significant association (*p* < 0.05) was observed between blood triglycerides, blood cholesterol, fecal triglycerides and fecal cholesterol (*p* < 0.05) (see Appendix A). Moreover, blood triglycerides showed a correlation with LP, CAT, GPx and GR (*p* < 0.05; see Appendix A), indicating that the acute hyperlipidemia model could compromise the endogenous antioxidant profile [4].

Previous studies have demonstrated that the polyphenol-rich fraction of the ACM fruit peel is not cytotoxic [10,21,30]. Therefore, given the exceptional antioxidant properties of CEAc and PFAc, they are ideal natural additives for preventing oxidation and offering additional health benefits, including reducing the risk of oxidative stress. Furthermore, the PFAc, which is particularly rich in proanthocyanidins, can serve as a natural colorant and flavor enhancer in food products. By utilizing the crude extract and polyphenols-rich fraction, there is potential for value-added applications of the fruit peel by-product, leading to waste reduction and promotion of sustainable production practices in the food industry.

## 4. Conclusions

In summary, the treatment with PAFc showed stronger antioxidant effects than that with CEAc. Overall, pre-treatment with ACM in the acute HLc mouse model improved TAC, as well as the activities of SOD, CAT, GPx, G6PD and GR, along with an increase in GSH content. Furthermore, the polyphenols-rich fraction from ACM fruit peel exhibited the ability to lower carbonyl content and LP status. Here, we demonstrated, for the first time, the antioxidant effects of the ACM fruit peel on the redox status of cardiac tissue after hyperlipidemia-induced oxidative stress. These recent findings provide support for further investigation into chronic hyperlipidemia models and clinical trials to explore the potential of ACM as a preventive and therapeutic management option for hyperlipidemia and related cardiovascular diseases.

## Figures and Tables

**Figure 1 foods-12-02097-f001:**
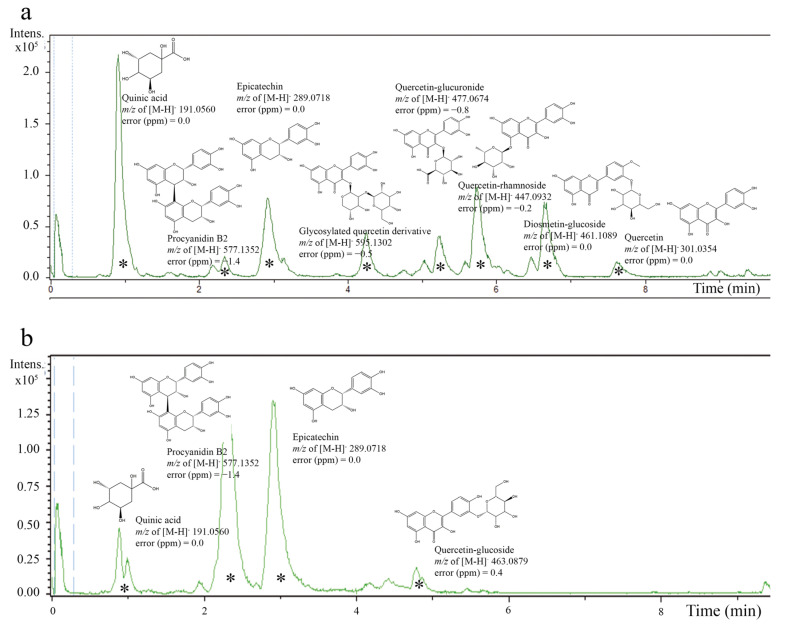
Electrospray ionization–mass spectrometry (ESI-MS) chromatogram of crude ethanol extract from *Annona crassiflora* fruit peel (CEAc) (**a**) and polyphenols-rich fraction from *Annona crassiflora* fruit peel (PFAc) (**b**) (negative mode). The compounds corresponding to major peaks were identified (*).

**Figure 2 foods-12-02097-f002:**
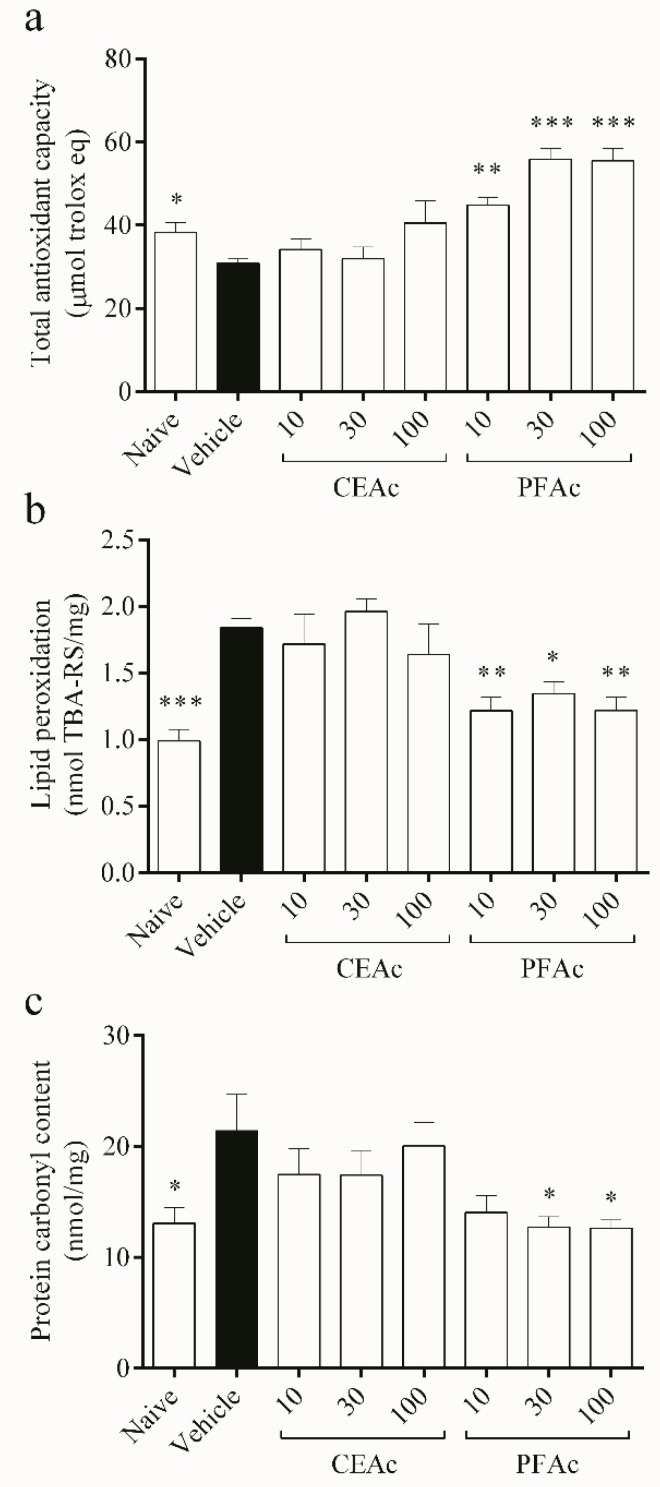
Assessment of overall antioxidant capacity (**a**), lipid peroxidation (**b**) and protein carbonylation (**c**) in the heart tissue of normolipidemic mice (naive), mice induced to be hyperlipidemic by tyloxapol but not receiving any treatment (referred to as “vehicle”), and mice induced to be hyperlipidemic by tyloxapol that were administered oral doses of 10, 30 or 100 mg/kg of the crude extract (CEAc) or polyphenols-rich fraction (PFAc) obtained from the fruit peel of *A. crassiflora*. The values are presented as mean ± standard error. * *p* < 0.05, ** *p* < 0.01 and *** *p* < 0.001 vs. vehicle (one-way ANOVA and Dunnett’s post-test).

**Figure 3 foods-12-02097-f003:**
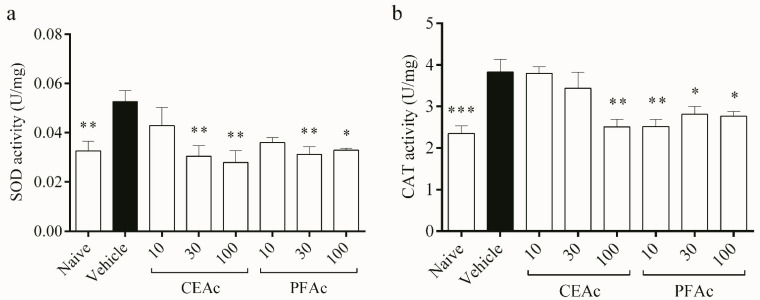
Assessment of catalase (CAT) (**a**) and superoxide dismutase (SOD) (**b**) in the heart tissue of normolipidemic mice (naive), mice induced to be hyperlipidemic by tyloxapol but not receiving any treatment (referred to as “vehicle”), and mice induced to be hyperlipidemic by tyloxapol that were administered oral doses of 10, 30 or 100 mg/kg of the crude extract (CEAc) or polyphenols-rich fraction (PFAc) obtained from the fruit peel of *A. crassiflora*. The values are presented as mean ± standard error. * *p* < 0.05, ** *p* < 0.01 and *** *p* < 0.001 vs. vehicle (one-way ANOVA and Dunnett’s post-test).

**Figure 4 foods-12-02097-f004:**
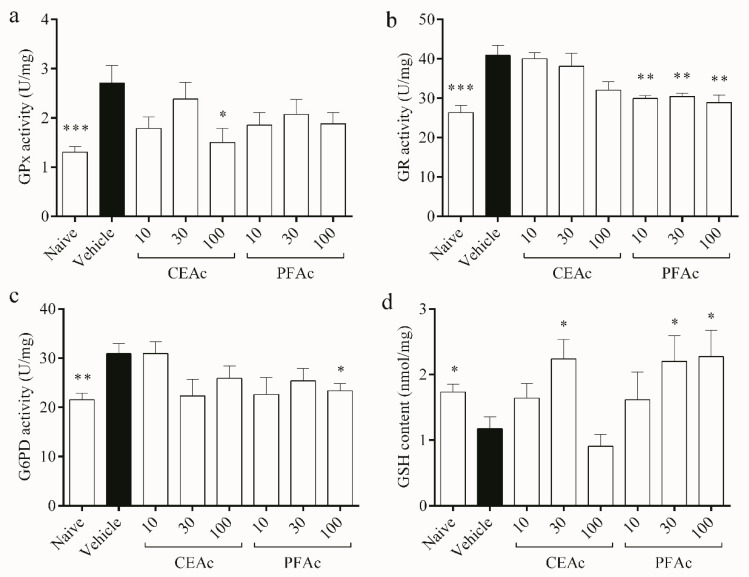
Assessment of glutathione peroxidase (GPx) (**a**), glutathione reductase (GR) (**b**), glucose 6-phosphate dehydrogenase (G6PD) (**c**) activities, and reduced glutathione (GSH) content (**d**) in the heart tissue of normolipidemic mice (naive), mice induced to be hyperlipidemic by tyloxapol but not receiving any treatment (referred to as “vehicle”), and mice induced to be hyperlipidemic by tyloxapol that were administered oral doses of 10, 30 or 100 mg/kg of the crude extract (CEAc) or polyphenols-rich fraction (PFAc) obtained from the fruit peel of *A. crassiflora*. The values are presented as mean ± standard error. * *p* < 0.05, ** *p* < 0.01 and *** *p* < 0.001 vs. vehicle (one-way ANOVA and Dunnett’s post-test).

## Data Availability

The data presented in this study are available in the article or Appendix A.

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
