# Peer review of "Annona crassiflora Mart. Fruit Peel Polyphenols Preserve Cardiac Antioxidant Defense and Reduce Oxidative Damage in Hyperlipidemic Mice"

_foods, 2023, doi:10.3390/foods12112097_

Round 1
Reviewer 1 Report
My questions are as follows:
1. Line88-92, according to the description “ CEAc (30.0 g) was solubilized in 225.0 mL of methanol: water (9:1) and subjected to a liquid-liquid partitioning.”, the solovent is methanol and water. However, “ethyl acetate fraction (polyphenols-rich fraction, named here as PFAc) were used for the tests.” . there is no method description about how to get the “ethyl acetate fraction” It is really confusing.
2. All the extract fractions were prepared by using chemical reagents. In order to make sure that the extract which was used to feed animals is safe and non-toxic, please provide information about if there is some chemical residue in the polyphenols extract.
3. Please add the qualitative and quantitative analysis of the polyphenols extract, and to help know which phenolic compound is responsible for the cardiac antioxidant defense
Author Response
- Line88-92, according to the description “ CEAc (30.0 g) was solubilized in 225.0 mL of methanol: water (9:1) and subjected to a liquid-liquid partitioning.”, the solovent is methanol and water. However, “ethyl acetate fraction (polyphenols-rich fraction, named here as PFAc) were used for the tests.” . there is no method description about how to get the “ethyl acetate fraction” It is really confusing.
Answer: We apologize for the absence of information about the liquid-liquid partition in the manuscript. To rectify this, we have included additional details on the solvents and crude extract fractionation procedures, as well as the methods used to remove solvents and water, as follows:
“Using a separation funnel, liquid-liquid partition was performed with solvents of increasing polarity: n-hexane, dichloromethane, ethyl acetate and n-butanol. For this process, four extractions were performed with approximately 200.0 mL of each solvent. Solvents from the fractions were completely removed by rotary evaporation under reduced pressure at 40°C (Bunchi Rotavapor R-210, Switzerland). The resulting fractions were frozen, lyophilized, and stored at -20 °C. The CEAc and ethyl acetate fraction, which is a polyphenol-rich fraction and referred to as PFAc, were used for the tests.”
- All the extract fractions were prepared by using chemical reagents. In order to make sure that the extract which was used to feed animals is safe and non-toxic, please provide information about if there is some chemical residue in the polyphenols extract.
Answer: We agree with the Reviewer that any solvent residue could be toxic and harmful to animals. In the present study, after ethanolic extraction and liquid-liquid partition, all solvents were completely removed from the fractions by rotary evaporation at 40°C under reduced pressure. This valuable information on the removal of solvents from the fractions has been added in the manuscript to make the reader clear about the safety of the materials administered to the animals.
- Please add the qualitative and quantitative analysis of the polyphenols extract, and to help know which phenolic compound is responsible for the cardiac antioxidant defense.
Answer: We appreciate the Reviewer's suggestion and, in response, we performed phytochemical prospecting to identify the primary bioactive compounds in both the crude extract (CEAc) and the polyphenol-rich fraction (PFAc) of the peel of A. crassiflora. To achieve this, we employed high-performance liquid chromatography and electrospray ionization mass spectrometry (HPLC-ESI-MS) for qualitative analysis. For quantitative investigations, we used the Folin-Ciocalteu assay to determine the total phenolic content of the samples. Given that the bioactive PFAc fraction is abundant in procyanidin B2 and epicatechin, which belong to the class of proanthocyanidins, we also used the acid vanillin method to quantify the proanthocyanidin content.

Reviewer 2 Report
Although fruit peels of Annona crassiflora Mart. (ACM) seem not to be eaten, indeed they are food-derived components that show antioxidant activity, being relevant for this Special Issue of FOODS.
To improve the interest of food specialists for this work, I suggest to include a topic that discusses how to apply the crude extract and polyphenols-rich fraction from AMC fruit peel in food products.
Some specific parts should be improved.
Material and methods:
2.1. Preparation and extraction of AMC polyphenols
The amount of solvents used in liquid liquid partition in this manuscript and in reference (Bioorganic Chemistry 69 (2016) 167-182) was not given. This means a difficulty for researchers who want to repeat the experimentation.
2.2. Animals and experimental design
The tyloxapol administration procedure was not clear.
Results and discussion:
3.1. Serum biochemical parameters
This part seems to be a summary of previous work. It might be skipped.
3.2. Oxidative stress markers and antioxidant system analysis
With respect to depletion of GSH, the following:
The kinetic constants of GPx may be available and can be used to compare the concentration of GSH to them, which may confirm the depletion of GSH or not.
Author Response
Although fruit peels of Annona crassiflora Mart. (ACM) seem not to be eaten, indeed they are food-derived components that show antioxidant activity, being relevant for this Special Issue of FOODS.
To improve the interest of food specialists for this work, I suggest to include a topic that discusses how to apply the crude extract and polyphenols-rich fraction from AMC fruit peel in food products.
Answer: We appreciate the Reviewer's suggestion, and we included a topic that discusses the application of the crude extract and polyphenols-rich fraction from AMC fruit peel in food products, as follows:
“Previous studies have demonstrated that the polyphenol-rich fraction of ACM fruit peel is not cytotoxic [10, 22, 31]. Therefore, given the exceptional antioxidant properties of the crude extract and polyphenols-rich fraction, they are ideal natural additives for preventing oxidation and offering additional health benefits, including reducing the risk of oxidative stress. Furthermore, the PFAc, which is particularly rich in proanthocyanidins, can serve as a natural colorant and flavor enhancer in food products. By utilizing the crude extract and polyphenols-rich fraction, there is poten-tial for value-added applications of the fruit peel by-product, leading to waste reduc-tion and promoting sustainable production practices in the food industry.”
Some specific parts should be improved.
Material and methods:
2.1. Preparation and extraction of AMC polyphenols
The amount of solvents used in liquid liquid partition in this manuscript and in reference (Bioorganic Chemistry 69 (2016) 167-182) was not given. This means a difficulty for researchers who want to repeat the experimentation.
Answer: We apologize for the absence of information about the liquid-liquid partition in the manuscript. To rectify this, we have included additional details on the solvents and crude extract fractionation procedures, as well as the methods used to remove solvents and water, as follows:
“Using a separation funnel, liquid-liquid partition was performed with solvents of increasing polarity: n-hexane, dichloromethane, ethyl acetate and n-butanol. For this process, four extractions were performed with approximately 200.0 mL of each solvent. Solvents from the fractions were completely removed by rotary evaporation under reduced pressure at 40°C (Bunchi Rotavapor R-210, Switzerland). The resulting fractions were frozen, lyophilized, and stored at -20 °C. The CEAc and ethyl acetate fraction, which is a polyphenol-rich fraction and referred to as PFAc, were used for the tests.”
2.2. Animals and experimental design
The tyloxapol administration procedure was not clear.
Answer: In fact, the process of inducing hyperlipidemia with tyloxapol was not well-defined. Therefore, we rewrote the methodological procedure for inducing hyperlipidemia with tyloxapol and its administration, as follows:
“Initially, the animals were given CEAc or PFAc (diluted in filtered water) daily for 12 days, while the naïve and HLc vehicle groups received filtered water. On the thir-teenth day, an intraperitoneal injection of 400 mg/kg of Triton WR-1339 (tyloxapol) was administered to the animals to induce a dyslipidemic condition, except for the mice in the naïve group [12]. CEAc and PFAc were orally administered for the last time to the treated groups after 24 h. After 48 h of tyloxapol (Tx) administration, the animals were euthanized intraperitoneally with ketamine (80 mg/kg) and xylazine (10 mg/kg) (diluted in sodium phosphate buffer, pH 7.3).”
Results and:
3.1. Serum discussion biochemical parameters
This part seems to be a summary of previous work. It might be skipped.
Answer: Serum biochemical parameters were analyzed in a previous study; however, they were important results that were used for correlation analyses with oxidative and antioxidant parameters in the present study. We have considered the reviewer's suggestion and restructured the results and discussion section by removing the separate topic on serum biochemical parameters and incorporating this information into the body of the discussion text.
3.2. Oxidative stress markers and antioxidant system analysis
With respect to depletion of GSH, the following:
The kinetic constants of GPx may be available and can be used to compare the concentration of GSH to them, which may confirm the depletion of GSH or not.
Answer: We agree with the Reviewer and through the evaluation of kinetic parameters involving the GPx enzyme and its substrate GSH, it was possible to complement the information about the depletion of GSH, as follows:
“This increase in GPx activity may have led to a depletion of GSH in the hyperlipidemic animals, as GSH servers as a substrate for GPx during the detoxification of H2O2 [29]. This can be confirmed through the kinetic parameters obtained from GPx. The naive group had a Km of approximately 0.1 mM, while the hyperlipidemic group had a Km of approximately 0.029 mM. This indicates that GPx in hyperlipidemic animals has a greater affinity for GSH than non-hyperlipidemic animals. Therefore, it confirms glutathione depletion by increasing GPx activity.”

Round 2
Reviewer 1 Report
According to the description shown in section 2.1 “Preparation and extraction of ACM polyphenols, we can find that the extract tested here is a crude polyphenols extract. Of course we all know that the solvents could be removed by rotary evaporation at 40°C under reduced pressure, but the residue of metal ions involved in the methanol and ethyl acetate is kept in the extract. Since the extract is the subject of this study, I advise to show some supplementary materials about the detail information about the composition of the extract to illustrate what were fed to the animals.
Author Response
According to the description shown in section 2.1 “Preparation and extraction of ACM polyphenols, we can find that the extract tested here is a crude polyphenols extract. Of course we all know that the solvents could be removed by rotary evaporation at 40°C under reduced pressure, but the residue of metal ions involved in the methanol and ethyl acetate is kept in the extract. Since the extract is the subject of this study, I advise to show some supplementary materials about the detail information about the composition of the extract to illustrate what were fed to the animals.
Anwer: First of all, we would like to thank you for your suggestion in the Reviewer's comments. We understand the concern regarding any potential metal residue that could be toxic to animals. However, we want to clarify that both methanol and ethyl acetate solvents used in the extraction and fractionation process were specific for HPLC, with a degree of purity of ≥99.9%. The following specifications were maintained during the process: ≤0.0005% non-volatile matter, ≤1 ppb fluorescence (quinine) at 254 nm, ≤1 ppb fluorescence (quinine) at 365 nm, ≤0.005% free acid (as CH3COOH) and <0.03% water (Karl Fischer), with a maximum of 10 ppm residue after evaporation. We agree that these specifications should have been included in the methodology section to reinforce the purity of the solvents and the safety of the fractions obtained after the rotaevaporation and lyophilization processes. Therefore, we have added these specifications and details to the manuscript to address this concern.
The manuscript now includes a mass spectrometry analysis that shows high-resolution spectra of molecular ions, without the presence of ions corresponding to possible impurities that could be present in the fractions. Additionally, the same fractions have already been used in other in vivo experiments involving rats and mice, which demonstrated that they are not toxic to animals. Moreover, we performed cytotoxicity analysis on the polyphenol-rich fraction, which demonstrated that it maintained cell viability. We have added this information to the manuscript to ensure the negligible presence of impurities and toxic residues in the samples.